# Targeting Alternative Splicing as a Potential Therapy for Episodic Ataxia Type 2

**DOI:** 10.3390/biomedicines8090332

**Published:** 2020-09-05

**Authors:** Fanny Jaudon, Simona Baldassari, Ilaria Musante, Agnes Thalhammer, Federico Zara, Lorenzo A. Cingolani

**Affiliations:** 1Department of Life Sciences, University of Trieste, 34127 Trieste, Italy; fjaudon@units.it; 2Unit of Medical Genetics, IRCCS Istituto Giannina Gaslini, 16147 Genoa, Italy; simonabaldassari@gaslini.org (S.B.); ilaria.musante@unige.it (I.M.); FedericoZara@gaslini.org (F.Z.); 3Department of Neurosciences, Rehabilitation, Ophthalmology, Genetics, Maternal and Child Health (DINOGMI), University of Genoa, 16126 Genoa, Italy; 4Center for Synaptic Neuroscience and Technology, Istituto Italiano di Tecnologia (IIT), 16132 Genoa, Italy; athalham@sissa.it; 5IRCCS Ospedale Policlinico San Martino, 16132 Genoa, Italy

**Keywords:** episodic ataxia type 2, P/Q-type Ca^2+^ channels, alternative splicing, antisense oligonucleotides, SMaRT, CRISPR/Cas9

## Abstract

Episodic ataxia type 2 (EA2) is an autosomal dominant neurological disorder characterized by paroxysmal attacks of ataxia, vertigo, and nausea that usually last hours to days. It is caused by loss-of-function mutations in *CACNA1A*, the gene encoding the pore-forming α_1_ subunit of P/Q-type voltage-gated Ca^2+^ channels. Although pharmacological treatments, such as acetazolamide and 4-aminopyridine, exist for EA2, they do not reduce or control the symptoms in all patients. *CACNA1A* is heavily spliced and some of the identified EA2 mutations are predicted to disrupt selective isoforms of this gene. Modulating splicing of *CACNA1A* may therefore represent a promising new strategy to develop improved EA2 therapies. Because RNA splicing is dysregulated in many other genetic diseases, several tools, such as antisense oligonucleotides, *trans*-splicing, and CRISPR-based strategies, have been developed for medical purposes. Here, we review splicing-based strategies used for genetic disorders, including those for Duchenne muscular dystrophy, spinal muscular dystrophy, and frontotemporal dementia with Parkinsonism linked to chromosome 17, and discuss their potential applicability to EA2.

## 1. Introduction

Episodic ataxia type 2 (EA2) is an autosomal dominant neurological disorder characterized by recurrent disabling attacks of imbalance, vertigo, nausea and ataxia, typically lasting hours to days [1,2]. Symptoms may also include fatigue, migraine headaches, and visual disturbances. Nystagmus commonly occurs between attacks and progressive cerebellar atrophy can also be observed. The frequency of attacks ranges from once a year to four times a week. Ataxic episodes can be triggered by emotional stress, physical exercise, alcohol intake, or fever. Although late-onset cases have been reported, the onset is typically in childhood or early adolescence [3,4]. Acetazolamide and 4-aminopyridine can reduce or control the symptoms in some patients. However, treatments are sometimes discontinued because the drugs are either no longer effective or because the patients develop adverse effects to them [5].

EA2 is caused by mutations in *CACNA1A*, the gene encoding the pore-forming α_1_ subunit of P/Q-type voltage-gated Ca^2+^ channels (VGCCs; Ca_V_2.1) [6,7]. These channels are enriched in the cerebellum where they control neurotransmitter release in cooperation with N-type (Ca_V_2.2) and R-type (Ca_V_2.3) VGCCs. P/Q-type channels are however more efficient than the other two types of VGCCs in supporting synaptic transmission at mature synapses, partly because they are more tightly coupled to the neurotransmitter release machinery [8,9,10]. The *CACNA1A* gene, located on chromosome 19p13, contains 47 exons, many of which are subject to alternative splicing (AS). As a consequence, the total number of *CACNA1A* splice isoforms is estimated to be in the order of thousands. Moreover, functional studies indicate that some of them exhibit differential properties and expression patterns [7,11]. This creates a large molecular variability that is thought to optimize Ca^2+^ signaling to specific cellular tasks.

Over 100 different mutations can cause EA2 (Figure 1; Table 1). Some of them are nonsense loss-of-function mutations that disrupt the open reading frame of *CACNA1A* and result in truncated channels generating little or no current. EA2 mutations are found in all regions of the channel but most of them reside in the pore loop region (Figure 1) [4,6,12]. Interestingly, some mutations are predicted to induce aberrant splicing.

For example, G to A substitutions at position +1 of the donor splice sites in exons 6, 11, 21, 24, 26, and 27 are all predicted to form aberrant mRNAs [6,13,14,15]. Likewise, mutations G to A at position +5 of the donor splice site in exon 4, G to A at position −1 in exon 7, T to C at position +2 of the donor splice site in exon 19 and a four-base pair deletion at position +4 of the donor splice site in exon 41 also impair splicing [4,14,16,17]. Further, the insertion of a T at position +3 of intron 24 creates a cryptic splice donor site within exon 24, leading to an aberrant splicing event [18]. While all the above mutations affect constitutive splicing, defects in AS are also important for EA2. Specifically, an A to G substitution at position −2 of the acceptor splice site in the mutually exclusive exon 37a has been found in EA2 patients [19]. This mutation likely impairs the insertion in the final transcript of exon37a while not affecting the inclusion of its mutually exclusive exon 37b.

The mutually exclusive exons 37a and 37b produce two major isoforms of the channel, Ca_V_2.1[EFa] and Ca_V_2.1[EFb], which diverge in an EF-hand-like domain located in the proximal intracellular C-terminus [20,21,22]. The ratio between the synaptic expression of Ca_V_2.1[EFa] and Ca_V_2.1[EFb] is key to controlling synaptic efficacy: Ca_V_2.1[EFa] is tightly coupled to the neurotransmitter release machinery and support efficient vesicle release while Ca_V_2.1[EFb] is loosely coupled to the release machinery and participates in vesicle release only upon repetitive stimulations [7,23,24]. The developmental expression pattern of the two isoforms is also markedly different. Whereas, for Ca_V_2.1[EFb], the expression levels do not undergo major developmental changes, being high in most brain regions from the early stages of postnatal development, for Ca_V_2.1[EFa], the expression levels build up during synapse maturation, presumably contributing to the developmental increase in neurotransmitter release efficiency. As a result, both splice isoforms are expressed at similar levels in most regions of the adult brain [20,21,22,23,25]. In addition to the aforementioned mutation impairing inclusion of exon 37a [19], four loss-of-function mutations have been identified within exon 37a in four unrelated families [4,26], while none has been found, to date, in exon 37b (Figure 1; Table 1). The presence of five mutations impairing selectively Ca_V_2.1[EFa] may therefore suggest a major role of this splice isoform in EA2.

Altogether, these findings indicate that aberrant splicing of *CACNA1A* may be an important pathogenic mechanism for EA2. Targeting RNA splicing may therefore be a promising approach for developing new EA2 therapies. After reviewing RNA splicing, AS, and splicing-based therapies developed for other diseases, we will discuss the potential benefits of splicing-based therapies for EA2.

## 2. Alternative Splicing

During the maturation of a pre-mRNA into an mRNA, non-coding introns are removed and exons are joined together, a process known as RNA splicing. By modulating which exons and how they are included in the mature mRNA, it is possible to generate multiple transcripts from a single gene, thereby producing protein isoforms with potentially different functions. This mechanism, known as AS, is a critical determinant of protein diversity. Importantly, more than 90% of human genes are alternatively spliced [61,62].

The splicing reaction is catalyzed by a large complex, the spliceosome, which is composed of five small nuclear RNAs (snRNAs) and more than 200 proteins [63]. snRNAs and proteins combine to form small nuclear ribonucleoproteins (snRNPs) that bind *cis*-regulatory elements on the pre-mRNA to catalyze the splicing reaction. The core splicing elements include a donor site at the 5′ end of the intron (the exon/intron junction) and an acceptor site at the 3′ end of the intron defined by a polypyrimidine tract and the actual intron/exon junction. The 3′ splice site is functionally coupled to the branch point, which is usually within 40 nucleotides upstream of the 3′ splice site.

Additional *cis*-regulatory sequences, known as exonic and intronic splicing enhancers/silencers (ESE/S and ISE/S), recruit specific RNA-binding proteins, such as serine/arginine-rich (SR) family proteins and heterogeneous nuclear ribonucleoprotein particles (hnRNPs) [64]. These additional elements facilitate or inhibit the assembly of the spliceosome, thereby regulating RNA splicing and making AS possible [65,66,67].

The most common type of AS is exon skipping, also known as cassette exon, in which a whole exon can be either included in or spliced out of the mRNA. Other types of AS comprise intron retention, in which an intron is included in the mature mRNA, mutually exclusive exons, whereby the transcript contains only one of two (or more) possible exons, and alternative 3′ and 5′ splice site selection, leading to alternate inclusion or exclusion of part of an exon [68,69].

Aberrant splicing is associated with a growing number of diseases [70,71,72]. Mutations affecting the core splice sites generally result in exon skipping or intron retention, which disrupt the reading frame, thus generating truncated non-functional proteins with potentially pathological consequences. In addition, also defects in AS, by mutations in splicing enhancers/silencers or in *trans*-acting elements, can be pathological. Several therapeutic approaches aiming at correcting aberrant splicing have therefore been developed, as detailed below.

## 3. Splicing Modulating Therapies

### 3.1. Antisense Oligonucleotides (AONs)

The antisense oligonucleotide (AON) approach is based on short oligonucleotides that bind the pre-mRNA and modify gene expression by either promoting mRNA degradation or interfering with the splicing process [73,74,75,76]. RNA-degrading AONs typically contain a phosphorothioate (PS) backbone modification that increases their stability and promotes the recruitment of the RNase H to induce cleavage of the target RNA. Other modifications, in the phosphodiester backbone and at the 2′ position of the sugar moiety, are also possible for improved AON properties [76,77]. For example, the morpholino backbone modification promotes highly stable RNA complementarity without inducing degradation of the target sequence. Likewise, 2′-O-methyl (2′-O-Me) and 2′-O-methoxyethyl (2′MOE) modifications enhance AON binding to the mRNA while inhibiting RNase H cleavage. AONs with modifications that protect from RNA degradation can regulate splicing if they are designed to recognize intron/exon junctions or splicing regulatory elements [74,78,79,80] because they prevent binding of splicing factors, thereby inducing exon skipping or forcing the inclusion of alternatively spliced exons [74,81]. Several AON-based therapeutic approaches, some of which are currently in clinical trials, have been developed for pathologies in which splicing is defective [82], as further discussed below.

#### 3.1.1. AON-Mediated Exon Inclusion

One disease for which AONs have been used to promote exon inclusion is spinal muscular atrophy (SMA; Table 2). SMA is a recessive, progressive, neuromuscular disorder caused by mutations in the survival motor neuron 1 (*SMN1*) gene [83]. It results in motor neuron degeneration and subsequent muscular atrophy leading to paralysis and respiratory defects, ultimately reducing the life expectancy of patients. Depending on the age of onset and level of motor function, SMA can be classified into four different subtypes SMA types I–IV [84,85,86,87]. Humans have a paralogous of *SMN1*, referred to as *SMN2*, which differs from *SMN1* in only few bases. Interestingly, the severity of SMA is inversely correlated to *SMN2* expression levels [88,89]. *SMN2* fails however to fully compensate for the lack of *SMN1* because of a C-to-T substitution at position 6 of exon 7, which disrupts an exonic splicing enhancer, thus promoting skipping of this exon with consequent formation of an unstable protein [86,90,91]. Instead of trying to correct the SMN mutations in *SMN1*, researchers have designed AONs targeting an intronic splicing silencer (ISS) in *SMN2* to prevent the binding of negative splicing factors, thus favoring exon 7 inclusion in the *SMN2* mRNA. This increases the amount of SMN protein in human fibroblasts and in the mouse brain [92,93,94]. Following these results, nusinersen, a 2′-O-methyl-modified ribose AON with a full-length phosphorothioate backbone (2′OMePS), which increases the inclusion of exon 7 in the *SMN2* mRNA, has been delivered in patients using intrathecal injections. These clinical trials indicated that the drug is generally well tolerated and effective in alleviating SMA symptoms and improving motor functions [95,96,97,98,99]. In addition, investigations of neuronal tissue in treated deceased patients have revealed a two- to six-fold increase in *SMN2* transcripts containing exon 7, as well as an increase in SMN protein, as compared to untreated patients [100]. Following these positive outcomes, nusinersen has been approved as SMA treatment by the US Food and Drug Administration (FDA) in 2016 and by the European Medicines Agency in 2017 [100]. Ongoing studies are evaluating the long-term safety and tolerability of nusinersen and its effects on presymptomatic patients [101].

Interestingly, combining different AONs can have synergic effect. For example, *SMN-AS1*, a long noncoding RNA (lncRNA) arising from the antisense strand of the *SMN* genes, represses SMN expression. AONs degrading *SMN-AS1* increase SMN expression in patient-derived cells and in the mouse brain. When lncRNA-targeting and splice-switching AONs are combined there is an even larger increase in SMN levels containing exon 7 [102].

#### 3.1.2. AON-Mediated Exon Skipping

AON-based approaches have been used as therapeutic strategy also in Duchenne muscular dystrophy (DMD; Table 2). This is a genetic X-linked recessive disease caused by mutations in the *DMD* gene that encode the cytoskeletal protein dystrophin. Patients manifest initial symptoms, such as muscular weakness and walking abnormalities, within the first years of life, and lose the ability to walk by the age of 12–14 years. Children with DMD eventually develop also cardiomyopathies and breathing problems due to weakness in respiratory muscles, which eventually lead to death in late adolescence [103,104]. Among inherited pediatric muscular dystrophies, DMD is the most common, with an incidence of 1/3500 male births [105]. Most DMD patients lack a functional dystrophin protein as a consequence of point mutations or frame-shifting deletions/insertions in the *DMD* gene [106,107]. By contrast, mutations leading to a partially functioning dystrophin protein are associated with the milder Becker muscular dystrophy (BMD) [108].

Over the last years, researchers have developed AON-mediated approaches that promote exon skipping of specific exons in order to restore the open reading frame of *DMD*. Two main types of compounds have been investigated: drisapersen, a 2′OMePS, and eteplirsen, a phosphorodiamidate antisense morpholino oligonucleotide (PMO), both of which promote skipping of exon 51 by binding to exon 51-splicing enhancers on the DMD pre-mRNA [109,110]. Skipping of this exon restores the open reading frame, thus promoting the translation of an internally truncated dystrophin protein that is capable of converting a severe DMD into a milder BMD [110,111,112]. Despite initial encouraging results with intramuscular injections of drisapersen in four DMD patients [110,111,112], phase II and phase III clinical trials have failed to demonstrate a significant clinical benefit or a clear increase in dystrophin expression [113,114]. Drisapersen has therefore not received FDA approval [73,114]. By contrast, eteplirsen was approved by the FDA in September 2016 for DMD patients harboring a mutation that can be compensated by exon 51 skipping (which occurs in about 13% of the DMD population) [113]. The extent of eteplirsen clinical benefits are however still debated and larger clinical trials are being conducted [115].

Another class of promising drugs for treating DMD are tricyclo-DNAs (tcDNA). These are conformationally constrained AONs specially designed to limit torsional flexibility of the sugar backbone in order to stabilize tcDNA/RNA heteroduplexes [115]. Their efficacy has recently been demonstrated in mouse models of DMD, where they induced consistently higher levels of exon skipping and rescue of dystrophin protein levels than 2′OMePSs and PMOs [116,117]. Toxicological and tolerance studies are currently being conducted to evaluate the possibility of clinical applications.

Although deletions flanking the exon 51 are the most common cause of DMD [118], multiple genomic defects affecting different regions of the *DMD* gene have been described in DMD. To circumvent this issue, ongoing studies aim at developing multiple exon skipping strategies [115].

#### 3.1.3. AON-Mediated Masking of Cryptic Splice Sites

In addition to promoting inclusion or skipping of exons, AONs can block the recognition of cryptic splice sites. This approach has for example been used successfully to correct the genetic defects of ataxia-telangiecatasia and Hutchinson-Gilford progeria syndrome (Table 2).

Ataxia-telangiecatasia is a rare, neurodegenerative, autosomal recessive disease characterized by cerebellar degeneration, telangiectasia (spider veins), immunodeficiency, cancer susceptibility and radiation sensitivity. It is caused by mutations in the *ATM* gene, which encodes the ubiquitously expressed serine/threonine kinase ATM, involved in controlling cell cycle checkpoints and in repairing damaged DNA [119]. About 50% of identified mutations affect splicing and 30–40% of these splicing mutations create new cryptic sites or interfere with splice regulatory elements [120,121,122]. Customized antisense morpholino oligonucleotides (AMOs) were designed to target three different mutations in *ATM*: (i) 7865C3T (A2622V), which creates a new 5′ splice site within exon 55, resulting in a deletion of the last 64 nt; (ii) 513C3T (Y171Y), which activates a cryptic 3′ splice site within exon 8, inducing deletion of the first 22 nt of exon 8; (iii) IVS28-159 A3G in intron 28, which activates cryptic 5′ and 3′ splice sites, resulting in insertion of a 112 nt segment from intron 28. The AMOs were effective in masking the pathogenic cryptic splice sites, thus restoring correct splicing and increasing functional ATM protein levels [123]. Further, coupling ATM-targeting AMOs to arginine-rich cell-penetrating peptides improved their capacity to cross the blood brain barrier and hereby their efficiency in correcting splicing [124]. Similar AMO-mediated strategies were also used effectively to correct splicing defects induced by other *ATM* mutations [120,121], thus highlighting the general applicability of this approach for ataxia-telangiecatasia.

Hutchinson-Gilford progeria syndrome is a rare autosomal genetic disorder whose main feature is premature aging [125]. The disease is caused by mutations in the *LMNA* gene, which encodes two nucleophilic A-type lamins, lamin A and lamin C. The most common mutation is a de novo substitution c.1824C>T in exon 11, which has the double effect of creating a cryptic splice site and disrupting a probable exon splicing enhancer [126,127]. The use of the cryptic splice site produces a truncated mRNA lacking the last 150 nucleotides of exon 11, thereby resulting in a muted protein, which is called progerin. AMOs designed to prevent access of the splicing machinery to the cryptic splice site restored normal splicing and rescued nuclear morphology in fibroblasts from patients [128]. Likewise, treating knock-in mice harboring the c.1827C>T mutation (equivalent to the c.1824C>T mutation in humans) with a combination of AMOs targeting the cryptic splice site in exon 11 and a 5′ splice site in exon 10 reduced the accumulation of progerin, thereby extending the life span of knock-in mice [129].

### 3.2. Spliceosome-Mediated RNA Trans-Splicing (SMaRT)

A different approach for reprogramming RNA splicing is spliceosome-mediated RNA *trans*-splicing (SMaRT). It consists of a *trans*-splicing reaction between an exogenous pre-*trans*-splicing RNA molecule (PTM) and a target endogenous pre-mRNA that induces the formation of a chimeric mRNA [130]. A typical PTM is composed of a binding domain, allowing the recognition of the target mRNA, intronic splicing elements, necessary for the splicing reaction, and the coding sequence to be substituted [131]. Depending on the orientation of the PTM, it is possible to replace 5′-, 3′- or internal gene portions. Although not yet used in clinical applications, this strategy has been successful in in vitro and in vivo models of cystic fibrosis (CF), frontotemporal dementia with parkinsonism linked to chromosome 17 (FTDP-17), and SMA (Table 2) [130,132].

CF is a common autosomal recessive disease due to loss-of-function mutations in a gene for a chloride channel, cystic fibrosis trans-membrane conductance regulator (CFTR). Deficiencies in CTFR cause the production of mucus secretions with higher viscosity, eventually obstructing pancreatic and respiratory tracts [133]. The most common mutation is ΔF508, a three base-pair deletion at codon 508 in exon 10, which results in deletion of a phenylalanine residue inducing misfolding and mislocalization of the chloride channel [134,135]. *Trans*-splicing with a PTM containing the coding sequence of exons 10–24 was effective in correcting this mutation at the mRNA level in both cultured human epithelial cells and a xenograft model of the disease [131,136].

SMaRT has also been used successfully to regulate AS in the *MAPT* gene, which encodes the microtubule associated protein tau, and whose mutation causes several pathologies including FTDP-17. *MAPT* contains 16 exons. AS of exons 2, 3, and 10 generates six different isoforms. In particular, exclusion or inclusion of exon 10 gives rise to isoforms containing three (3R) or four (4R) microtubule-binding repeats, respectively [137,138]. Normal function of the adult brain requires 3R and 4R isofoms to be expressed in similar amounts. Many FTDP-17-causing mutations promote inclusion of exon 10, thus increasing the levels of 4R tau, which alters axonal transport and hereby triggers FTDP-17 [139,140,141]. In a mouse model of tauopathies, it was possible to regulate the ratio between 4R and 3R tau isoforms by delivering, into the mouse brain, a PTM containing a binding domain complementary to the 3′ end of intron 9 followed by either exons 10–13 or exons 11–13 to produce full-length tau chimeric proteins with or without exon 10, respectively [142,143,144].

Likewise, a SMaRT strategy has been developed to compensate for the deficiency of functional SMN proteins in SMA [145,146,147,148]. Researchers have used a PTM containing the SMN1 exon 7 sequence, an optimized splice site and a sequence annealing to the SMN intron 6 to increase the expression of exon 7-containing SMN2 mRNAs in SMA fibroblasts [147]. Combining this PTM with an AON designed to inhibit the downstream splice site at exon 8 further increased the efficiency of *trans*-splicing [145] and could ameliorate the SMA phenotype of SMAΔ7 mice [146,148].

### 3.3. CRISPR-Based Approaches

In recent years, the CRISPR/Cas9 (clustered regularly interspaced short palindromic repeats/CRISPR-associated protein 9) system has emerged as a powerful tool for developing new gene editing therapies [149,150]. It relies on short guide RNA sequences (gRNAs) that bring the endonuclease Cas9 to specific genomic loci to induce double strand breaks (DSBs). Upon DSB, endogenous cell mechanisms mediate non-homologous end joining (NHEJ) DNA repair, resulting in insertion/deletion (InDels) mutations near the DSB site. Alternatively, in the presence of a homologous DNA donor template, CRISPR/Cas9 can be used to introduce precise mutations through homology-directed repair (HDR) [151]. HDR is however not as efficient as NHEJ and it is thought to occur primarily in the S and G2 phases of the cell cycle, making its use in post-mitotic cells, such as neurons, challenging [152]. Recently, CRISPR approaches have been adopted also to modulate RNA splicing.

An NHEJ-based CRISPR strategy has for example been developed to promote exon 7 inclusion in *SMN2*, thus compensating for *SMN1* deficiency in SMA (Table 2) [153]. As previously done using AONs [93], the authors did not try to correct the causative mutation in *SMN1*, rather they aimed at inactivating by NHEJ two intronic splicing silencers in intron 7 of *SMN2* (ISS-N1 and ISS + 100; Figure 2A). To this end, they expressed, in human iPSC-derived motor neurons, Cas9 and a gRNA targeting one or the other ISS. The inactivation of either ISS effectively promoted inclusion of exon 7 in *SMN2*, thus increasing the expression of full length SMN protein. Importantly, this was sufficient to inhibit degeneration of iPSCs-derived motor neurons carrying a SMA mutation. Likewise, co-injecting Cas9 mRNA and a gRNA targeting ISS-N1 into the zygote of a SMA mouse model was effective in improving motor functions and extending the lifespan of treated mice [153].

NHEJ-based approaches have also been used successfully to skip mutated exons in the *Dmd* gene of the mdx mouse model of DMD (Table 2). The mdx mouse harbors a nonsense mutation in exon 23 of *Dmd*, which generates a premature stop codon leading to the formation of a truncated non-functional dystrophin protein [154]. Expression of Cas9 together with two gRNAs targeting the mutated region and the 3′ or 5′ end of exon 23 induced skipping of this exon by NHEJ-mediated genome editing, thus restoring the open reading frame of *Dmd* [155]. Exon 23 could also be skipped by targeting Cas9 to the regions flanking exon 23 [156,157]. Both approaches were effective in increasing the expression of internally truncated dystrophin, thereby partially rescuing cardiac muscle deficiencies and improving muscle force in mdx mice [155,156,157].

A different but equally effective CRISPR-based strategy to regulate splicing and restore the open reading frame of *DMD* consists of fusing a cytidine deaminase to a mutated nuclease dead (d)Cas9. The chimeric protein can be targeted to specific DNA sequences by gRNAs where it mediates a C to T conversion without inducing a DSB [158,159]. Likewise, adenine base editors can mediate an A to G conversion in specific DNA regions when fused to dCas9 [160]. A CRISPR-guided cytidine deaminase was used to mutate the 5′ splice site of exon 50 of *DMD* in iPSCs derived from a DMD patient who lacked dystrophin because of a deletion of exon 51 (Table 2) [161]. Following this mutation, exon 50 was skipped, which restored the open reading frame, thus generating an internally truncated dystrophin capable of partially rescuing the phenotype of iPSC-derived cardiomyocytes (Figure 2B) [161]. By targeting specific splice sites, we believe that CRISPR-mediated base editing technologies may become a powerful tool also for regulating the usage of mutually exclusive exons, for favoring the utilization of alternative splice sites and for promoting intron retention.

Finally, the discovery of Cas nucleases that target and cleave RNA rather than DNA, such as those of the Cas13 superfamily, has led to the development of CRISPR therapeutic strategies based on RNA manipulations [162,163]. Targeting RNA presents the advantage of correcting disease-relevant transcripts without permanently altering the genome. A recent study has shown that this strategy is feasible and has therapeutic potential in FTDP-17. Expression of a nuclease dead (d)Cas13 together with three gRNAs targeting the splice acceptor and two putative ESEs in exon 10 of *MAPT* increased skipping of exon 10 by preventing binding of splicing factors. This effectively restored a normal 4R/3R tau ratio in iPSCs-derived cortical neurons derived from FTDP-17 patients (Table 2; Figure 2C) [164].

## 4. Conclusions: Potential Advantages of Splicing Therapies for EA2

Given the number of mutations predicted to disrupt the splicing of *CACNA1A* in EA2 [4,6,13,14,15,16,17,18,19,26], splicing-targeting approaches, as described in the previous paragraphs, may hold great promise for developing new EA2 therapies. Furthermore, skipping of mutated exons could be used in some cases to restore the open reading frame of *CACNA1A*, thereby making it possible to generate functional Ca_V_2.1 channels. These strategies would, however, require a specific design and an experimental validation for each targeted mutation.

As for SMA, where therapies target *SMN2* to compensate for a deficiency in *SMN1*, EA2 therapies could regulate AS in presynaptic VGCCs other than Ca_V_2.1 to compensate for a deficiency in this channel. At most central synapses, Ca_V_2.1 is aided by Ca_V_2.2 (aka: N-type channel; gene for the α_1_ subunit: *CACNA1B*) and Ca_V_2.3 (aka: R-type channel; gene for the α_1_ subunit: *CACNA1E*). *CACNA1B* and *CACNA1E* are homologous to *CACNA1A* (59.4% and 54.0% identity at the amino acidic level, respectively) and, like *CACNA1A*, are extensively spliced, with the splicing pattern often conserved across the three mammalian Ca_V_2 channels [7]. Notably, in mouse models of EA2, lack, reduced expression or functional impairment of Ca_V_2.1 induces a compensatory up-regulation in the expression of the other two synaptic VGCCs [165,166]. Despite this compensation, synaptic transmission remains severely compromised at many synapses, including those in the cerebellum, mostly because Ca_V_2.2 and Ca_V_2.3 channels maintain a loose and inefficient coupling configuration to the neurotransmitter release machinery [167,168,169,170,171]. In principle, this functional diversity could mainly be due to intrinsic differences between the three Ca_V_2 genes. Some interesting data indicate however that it is also extensively influenced by alternative splicing. For example, the mutually exclusive exons 37a (EFa) and 37b (EFb) are conserved across Ca_V_2.1, Ca_V_2.2 and Ca_V_2.3 channels [23,172]. While little is known about the functional relevance of this splicing event in Ca_V_2.3, the EFa isoforms of both Ca_V_2.1 and Ca_V_2.2 are more efficient than the respective EFb isoforms in supporting synaptic transmission [23,173,174]. The EFa isoform of Ca_V_2.2 is, however, enriched only in capsaicin-responsive nociceptors of dorsal root ganglia [175], where it mediates efficient synaptic transmission [173,174], whereas EFb is the predominant isoform of Ca_V_2.2 in most brain regions [175]. Molecular approaches designed to increase the relative abundance of the EFa isoform of Ca_V_2.2 (and possibly of Ca_V_2.3) in the brain could therefore help compensate for a Ca_V_2.1 deficiency. Importantly, such strategies that leverage a generalized up-regulation of ‘secondary’ presynaptic Ca^2+^ channels (Ca_V_2.2 and Ca_V_2.3) in EA2 rather than targeting individual EA2 mutations in the ‘primary’ presynaptic Ca^2+^ channel (Ca_V_2.1), of which there are more than 100, would have the advantage of being suitable for most EA2 patients.

## Figures and Tables

**Figure 1 biomedicines-08-00332-f001:**
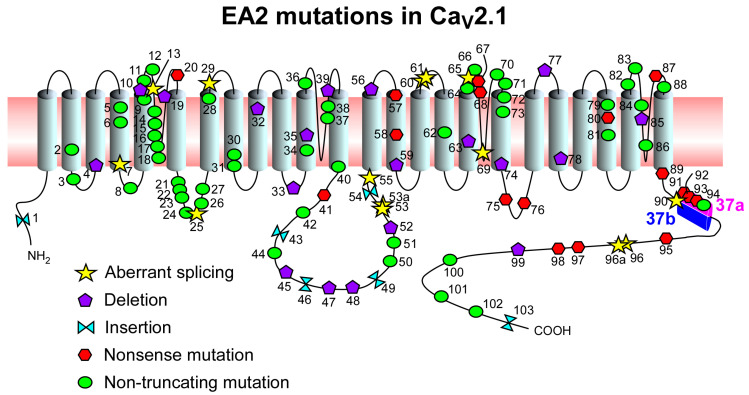
Scheme of human Ca_V_2.1 highlighting the position of 103 mutations causing EA2. Topology of Ca_V_2.1 as for Uniprot entry O00555. The two mutually exclusive exons 37a and 37b are shown in pink and blue, respectively. The 103 mutations are depicted in the scheme and listed in Table 1.

**Figure 2 biomedicines-08-00332-f002:**
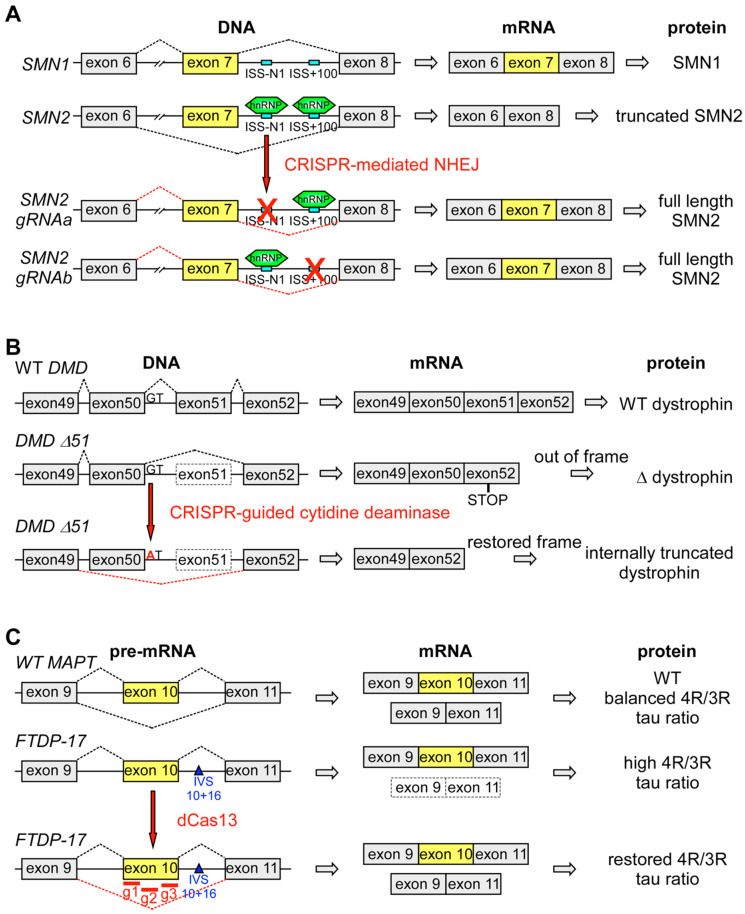
CRISPR-based splicing therapies (**A**) CRISPR-mediated non-homologous end joining (NHEJ) disrupts intronic splicing silencer (ISS) sites in intron 7 of *SMN2* preventing binding of hnRNPs, thus favoring inclusion of exon 7, normally not present in *SMN2*. This helps compensate for mutated *SMN1*. (**B**) Deletion of exon 51 of *DMD* leads to a frameshift resulting in a truncated nonfunctional dystrophin. CRISPR-guided cytidine deaminase mutates the 5′ splice site of exon 50, leading to skipping of exon 50, which restores the reading frame. The resulting internally truncated dystrophin is able to restore partially WT function. (**C**) AS of *MAPT* exon 10 leads to formation of 4R and 3R tau isoforms. The FTDP-17-associated IVS 10 + 16 mutation results in increased exon 10 inclusion and higher 4R tau levels. Expression of dCas13 together with three gRNAs targeting the exon 10 splice acceptor site and two putative exonic splicing enhancers (ESEs) promotes skipping of exon 10, thus restoring a balanced 4R/3R tau ratio. Schemes adapted from [149,161,164].

**Table 1 biomedicines-08-00332-t001:** List of Ca_V_2.1 mutations causing EA2.

No.	Amino Acid	DNA Mutation	References
1	p.(Ala56Serfs*20)	c.165dupA	[4]
2	p.Glu147Lys	c.439G>A	[27]
3	p.Gly162Val	c.485G>T	[28]
4	p.(Trp168Glyfs*10)	c.504delC	[4]
5	p.Arg192Trp	c.574C>T	[29]
6	p.Arg198Gln	c.593G>A	[30]
7		c.868+5G>A; possible aberrant splicing	[16]
8	p.Ser218Leu	c.653C>T	[30]
9	p.Tyr248Asn/Cys	c.742T>A/c.743A>G	[31,32]
10	p.Gly250Glufs*60	c.749delG	[12]
11	p.His253Tyr	c.1032C >T	[33]
12	p.(Cys256Arg)	c.1041T>C	[34]
13		c.983-1G>A; aberrant splicing	[4]
14	p.Arg279Cys	c.835C>T	[28]
15	p.Cys287Tyr	c.1096G>A	[1]
16	p.Gly293Arg	c.1152G>A	[35]
17	p.Gly297Arg	c.889G>A	[36]
18	p.Asp302Asn	c.904G>A	[28]
19	p.Thr310 fs*5	c.928_931delACTG	[28]
20	p.Trp320*	c.959G>A	[12]
21	p.Arg387Gly	c.1159C>G	[28]
22	p.Glu388Lys	c.1161G>A	[37]
23	p.(Leu389Phe)	c.1165C>T	[4]
24	p.Gly411Trp	c.1231G>T	[28]
25		c.1253+1G>A; probable aberrant splicing	[15]
26	p.Ala454Thr	c.1360G>A	[38]
27	p.Arg455Gln	c.1364G>A	[39]
28	p.(Thr501Met)	c.1502C>T	[4]
29		c.1557+1G>A; aberrant splicing	[13]
30	p.Glu533Lys	c.1597G>A	[40]
31	p.Gly540Arg	c.1618G>A	[41]
32	p.Val558Serfs*13	c.1672-1_1675delGGTTA	[28]
33	p.Leu600 fs*41	c.1799_1800delTC	[28]
34	p.Leu621Arg	c.2144T>G	[41]
35	p.Leu624Phe	c.1870-1873del	[13]
36	p.Gly638Asp	c.1913G>A	[42]
37	p.Trp670Cys	c.2010G>C	[30]
38	p.Gly677Glu	c.2030G>A	[31]
39	p.(Gln681Argfs*100)	c.2042_2043del	[33]
40	p.Ile712Val	c.2134A>G	[43]
41	p.Gln736*	c.2206C>T	[44]
42	p.(Met798Thr)	c.2393T>C	[4]
43	p.(Arg822Profs*246)	c.2464dupC	[4]
44	p.(Pro897Arg)	c.2690C>G	[4]
45	p.Gly939*	c.2816delG	[45]
46	p.Ser943Gln	c.2825+1insG	[46]
47	p.(Ala952Serfs*115)	c.2852_2861del	[4]
48	p.(Arg957Aspfs*113)	c.2867_2869del	[16]
49	p.Glu1004Argfs*66	c.3244+1insG	[1]
50	p.Glu998Gln	c.2992G>C	[31]
51	p.Gln1154*	c.3460C>T	[45]
52	p.Cys1178Pro	c.3531delC	[13]
53		c.3089+2T; possible aberrant splicing	[17]
53a		c.3102+2T; possible aberrant splicing	[30]
54		c.3603dupC	[30]
55		c.3977+1G>A; aberrant splicing	[14]
56	p.Pro1267Leu	c.4073delC	[6]
57	p.(Arg1278*)	c.3832C<T	[16]
58	p.Arg1281*	c.4077C>T	[1]
59	p.Glu1294Del	c.3871_3873delGAG	[31]
60		c.4270+1G>A; aberrant splicing	[6]
61		c.4001+3insT; cryptic splice donor site in exon 24	[18]
62	p.Arg1350Gln	c.4049G>A	[47]
63	p.Phe1394Leu	c.4182delC	[33]
64	p.Phe1404Cys	c.4486T>G	[48]
65		c.4261+1G>A; aberrant splicing	[13]
66	p.Arg1433Gln	c.4298G>A	[49]
67	p.Tyr1443*	c.4331C>G	[13]
68	p.Trp1451*	c.4588G>A	[1]
69		c.4636+1G>T; aberrant splicing	[1]
70	p.(Gly1483Arg)	c.4722G>A	[34]
71	p.Met1488_Ser1489del	c.4739_4744del	[34]
72	p.Phe1491Ser	c.4747T>C	[50]
73	p.(Val1494Ile)	c.4755G>A	[34]
74	p.Phe1503Del	c.4509-11delCTT	[13]
75	p.Arg1549*	c.4645C>T	[51]
76	p.Gln1561*	c.4963C>T	[52]
77	p.Tyr1594Del	c.4778-80delCTT	[13]
78	p.Val1620Ser	c.4854delG	[46]
79	p.Arg1666His/Gln	c.5260G>A/c.4991G>A	[53,54]
80	p.Arg1669*	c.5005C>T	[31]
81	p.(Arg1680Cys)	c.5038C>T	[4]
82	p.His1737Leu	c.5211A>T	[55]
83	p.Leu1749Pro	c.5246T>C	[28]
84	p.Arg1751Trp	c.5251C>T	[45]
85	p.Ser1753Cysfs*2	c.5253-2259_5403+1135del	[12]
86	p.Glu1757Lys	c.5271G>A	[56]
87	p.Arg1785*	c.5589C>T	[1]
88	p.Ser1799Leu	c.5396C>T	[57]
89	p.Arg1824*	c.5733C>T	[58]
90		c.IVS36+2T>C (c.4582+2T>C); possible aberrant splicing	[19]
91	p.Tyr1849*	c.5547T>A	[12]
92	p.Tyr1854*	c.5562C>G	[26]
93	p.Arg1858*	c.5571C>T	[26]
94	p.(Cys1870Arg)	c.5608T>C	[4]
95	p.(Glu1927*)	c.5779G>T	[4]
96		c.IVS41+(3–6)delGAGT (c.6068+(3-6)delGAGT); aberrant splicing	[18]
96a		c.6335+4delAGTG; aberrant splicing	[14]
97	p.Gln1986*	c.5956C>T	[31]
98	p.Gln2039*	c.6351C>T	[1]
99	p.Pro2058Leufs*69	c.6404delC	[1]
100	p.Arg2090Gln	c.6269G>A	[59]
101	p.(Arg2136Cys)	c.6681C>T	[34]
102	p.Pro2222Leu	c.6665C>T	[12]
103	p.2319Gln[n]	c.7191CAG[n]	[60]

**Note:** For most mutations, the numbering is as cited in the original publications. If not present in the original publication it refers to the translated region of the GeneBank access number AF004883.1. The asterisk (*) indicates a stop codon.

**Table 2 biomedicines-08-00332-t002:** Examples of Splicing-based Therapeutic Approaches.

Disease	Gene	Aberrant Splicing	Therapy	References
SMA	Mutated: *SMN1* Therapy: *SMN2*	Skipping of exon 7 due to C > T mutation at position 6 of exon 7	2′OMePS AONs targeting ISS-N1 to promote exon 7 inclusion in human fibroblasts and mouse brain	[92,93,94]
			Intrathecal injection of nusinersen to increase exon 7 inclusion in patients	[95,96,97,98,99,100]
			Splice switching and lncRNA-targeting AONs to promote inclusion of exon 7 in patient-derived cells and mouse brain	[102]
			*Trans*-splicing-mediated insertion of exon 7 in fibroblasts and SMAΔ7 mice	[145,146,147,148]
			CRISPR/Cas9-mediated disruption of intronic splicing silencers to enhance exon 7 inclusion in iPSC-derived motor neurons and mice	[153]
DMD	*DMD*	Formation of nonfunctional truncated dystrophin due to point mutations and frame-shifting deletions	2′OMePS- or PMO-mediated skipping of exon 51 in patients	[109,110,111,112,113,114,115]
			tcDNA-mediated exon skipping in mice	[116,117]
			Restoration of open reading frame by dual Cas9-mediated exon skipping in mdx mice	[155,156,157]
			dCas9-cytidine deaminase-mediated base editing to induce skipping of exon 50 and restore reading frame in iPSCs	[161]
FTDP-17	*MAPT*	Increased exon 10 inclusion	*Trans*-splicing to reverse aberrant exon 10 inclusion in mice	[142,143,144]
			dCas13 targeted to exon 10 splice sites on pre-mRNA to mediate exon exclusion in iPSCs-derived cortical neurons	[164]
Ataxia telangiectasia	*ATM*	Activation of cryptic splice sites	AMOs to mask cryptic splice sites and restore correct splicing in cell lines	[123,124]
Hutchinson-Gilford progeria syndrome	*LMNA*	Activation of an exonic cryptic donor splice site in exon 11	AMOs to block the recognition of cryptic splice sites in fibroblasts and mice	[128,129]
CF	*CFTR*	Three base-pair deletion in exon 10 causing deletion of F508	Correction of exon 10 by *trans*-splicing in epithelial cells and a xenograft model	[131,136]

**Abbreviations:** 2′OMePS: 2’-O-methyl-phosphorothioate RNA; AMO: Antisense morpholino oligonucleotide; AON: Antisense oligonucleotide; CF: cystic fibrosis; DMD: Duchenne muscular dystrophy; FTDP-17: Frontotemporal dementia with parkinsonism linked to chromosome 17; PMO: Phosphorodiamidate antisense morpholino oligonucleotide; SMA: Spinal muscular atrophy.

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
