# Peer review of "Targeting Alternative Splicing as a Potential Therapy for Episodic Ataxia Type 2"

_biomedicines, 2020, doi:10.3390/biomedicines8090332_

Round 1
Reviewer 1 Report
This review of Jaudon and colleagues provides a description of the molecular causes responsible for episodic ataxia 2 and an interesting overview of splicing-based strategies for the treatment of genetic disorders.
The following points should be addresses:
- Mutation nomenclature in the table associated with Figure 1 is not correct. Mutation nomenclature should be modified in accordance to the HGV guidelines: protein variants should be written in brackets when RNA nor protein have been analyzed, because they represent predictions. In addition, the three letters abbreviation for amino acids is preferred. For frameshift mutations, the change of the first amino acid affected should be indicated, followed by the indication of how many amino acids are left before the new stop codon appears. For example, for the first mutation (c.165dupA) the correct nomenclature is p.(Ala56Serfs*20). In addition, small insertions/deletions are also considered point mutations. In column two (DNA mutation), the authors may want to specify either the type of mutation (missense, frameshift, nonsense, splicing) or the exact mutation on cDNA level. But this should be done in the same way for all the mutations listed.
- The citation of reference 18 on page 2, line 19 is not correct. That paper (Wan et al., 2005) does not describe the activation of a cryptic splice donor site within exon 24. Wan and colleagues instead study the effects of the p.C287Y and p.G293R mutations at a molecular level.
- Reference 29 is indicated as “invalid citation”. Please check it out
- Paragraph 3.1.3 “AON-mediated masking of cryptic splice sites” the authors state that different AMOs were designed to target three different mutations in ATM (Page 8, line 6-11). How frequent are these three mutations in the population of patients with ataxia-telangiecatasia?
Author Response
Dear Biomedicines editorial team,
We would like to thank the Editors and Reviewers for taking the time to closely read our Review Article and for providing useful comments to improve sections of the manuscript. Below is a point-by-point response to the reviewers’ comments. Corrections in the revised manuscript were done using the ‘track changes’ option.
Reviewer 1
- Mutation nomenclature in the table associated with Figure 1 is not correct. Mutation nomenclature should be modified in accordance to the HGV guidelines: protein variants should be written in brackets when RNA nor protein have been analyzed, because they represent predictions. In addition, the three letters abbreviation for amino acids is preferred. For frameshift mutations, the change of the first amino acid affected should be indicated, followed by the indication of how many amino acids are left before the new stop codon appears. For example, for the first mutation (c.165dupA) the correct nomenclature is p.(Ala56Serfs*20). In addition, small insertions/deletions are also considered point mutations. In column two (DNA mutation), the authors may want to specify either the type of mutation (missense, frameshift, nonsense, splicing) or the exact mutation on cDNA level. But this should be done in the same way for all the mutations listed.
We thank the referee for his/her suggestion. We have now modified the table in Figure 1 according to the HGV guidelines.
- The citation of reference 18 on page 2, line 19 is not correct. That paper (Wan et al., 2005) does not describe the activation of a cryptic splice donor site within exon 24. Wan and colleagues instead study the effects of the p.C287Y and p.G293R mutations at a molecular level.
We have inserted the correct citation.
- Reference 29 is indicated as “invalid citation”. Please check it out
We have solved this formatting problem.
- Paragraph 3.1.3 “AON-mediated masking of cryptic splice sites” the authors state that different AMOs were designed to target three different mutations in ATM (Page 8, line 6-11). How frequent are these three mutations in the population of patients with ataxia-telangiecatasia?
We could not find the frequency of these three mutations. We however specify now the frequency, in ataxia-telangiecatasia, of splicing mutations (about 50%) and of those creating new cryptic splice cite and/or interfering with splice regulatory elements (30-40% of the splicing mutations).
Reviewer 2 Report
This review is comprehensive and can be considered for publication.
However, I have some concerns:
- Abstract and Introduction: " these drugs do not prevent the progression of the disease" In fact however EA2 is not really a progressive disease in most patients
- Abstract:"brain disorder" --> "neurological disorder" sounds more appropriate
- It's not clear what is the genetic similarity between EA2 and SMA: Humans have a paralogous of SMN1, referred to as SMN2. This is not true for the EA2 associated-gene. Therefore, I can't see how the results obtained on SMA may be transferred to EA2. The other disorders which have been discussed appear to have rather specific genetic mechanisms as well. The point of the applicability to EA2 of these findings should really be clarified and expanded.
Author Response
Dear Biomedicines editorial team,
We would like to thank the Editors and Reviewers for taking the time to closely read our Review Article and for providing useful comments to improve sections of the manuscript. Below is a point-by-point response to the reviewers’ comments. Corrections in the revised manuscript were done using the ‘track changes’ option.
Review 2
- Abstract and Introduction: " these drugs do not prevent the progression of the disease" In fact however EA2 is not really a progressive disease in most patients
We agree with the referee. We have removed this sentence.
- Abstract:"brain disorder" --> "neurological disorder" sounds more appropriate
We have introduced this change in the abstract and in the main text.
- It's not clear what is the genetic similarity between EA2 and SMA: Humans have a paralogous of SMN1, referred to as SMN2. This is not true for the EA2 associated-gene. Therefore, I can't see how the results obtained on SMA may be transferred to EA2. The other disorders which have been discussed appear to have rather specific genetic mechanisms as well. The point of the applicability to EA2 of these findings should really be clarified and expanded.
We thank the referee for raising this point. We have extensively rewritten the last paragraph of the review to better discuss how the homology between the three CaV2 channels and their similar splicing patterns could be employed to compensate for deficiencies in CaV2.1.
Round 2
Reviewer 1 Report
The authours have addressed my concerns
Reviewer 2 Report
This emended version of the manuscript may be suitable for publication